# Managing the Mental Health of Healthcare Professionals in Times of Crisis: The Aruban COVID-19 Experience

**DOI:** 10.3390/healthcare10071263

**Published:** 2022-07-07

**Authors:** Veronika Duwel, Jaclyn M. L. de Kort, Shailing S. Jacobs, Robert M. Dennert, Jamiu O. Busari

**Affiliations:** 1Faculty of Health and Life Sciences, Maastricht University, P.O. Box 616, 6200 MD Maastricht, The Netherlands; veronikaduwel@gmail.com; 2Department of Internal Medicine, Horacio E. Oduber Hospital, Oranjestad, Aruba; j.dekort@hoharuba.com; 3Department of General Medicine, Horacio E. Oduber Hospital, Oranjestad, Aruba; s.jacobs@hoharuba.com; 4Department of Cardiology, Horacio E. Oduber Hospital, Oranjestad, Aruba; r.dennert@hoharuba.com; 5Department of Educational Development and Research, Faculty of Health, Medicine and Life Sciences, Maastricht University, P.O. Box 616, 6200 MD Maastricht, The Netherlands; 6Department of Paediatrics, Horacio Oduber Hospital, Oranjestad, Aruba

**Keywords:** health workforce, multidisciplinary, low resource setting, mental health, COVID-19

## Abstract

Hospital workers in Aruba have been facing an increased demand for healthcare in the unique setting of a Small Island Developing State (SIDS). This study assessed the impact of the first wave of the SARS-CoV-2 pandemic on the mental health of staff at the major hospital in Aruba, examining the differences between employee groups, with the goal of providing recommendations for targeted support and coping strategies in future crises in a small island setting. **Patients and methods:** In a mixed-method cohort design, Dr. Horacio E. Oduber Hospital staff were asked to complete a 25-item questionnaire about their concerns and worries, organization of work, and general wellbeing; 24% of the hospital staff filled in the questionnaire (mean age 41 ± 11 years, 79% female). Alongside the needs assessment questionnaire, six focus groups were established to explore staff feelings on specific measures taken by hospital management during the COVID-19 crisis. **Results:** Questionnaire analysis (*n* = 231) revealed employees’ concerns about infecting their relatives and their financial stability. In particular, nurses were significantly more concerned than other staff groups. In the wellbeing section of the questionnaire, items regarding future security scored poorest, alongside increased levels of tiredness and nervousness. Focus groups discussions revealed frustrations of the hospital staff with the foreign staff brought in to help during the crisis and a need for better leadership and communication practices from hospital management. **Conclusions:** Comprehensive and holistic approaches should be implemented by the hospital management to prevent occupational burnout and demoralized work ethics and further emotional exhaustion.

## 1. Introduction

The COVID-19 pandemic, like any major health crisis, has had a significant impact on the lives of people all over the world. Spanning all continents, hardly any country has been spared the devastation of the COVID-19 pandemic that has claimed the lives of millions. In addition to the loss of lives, there has also been an existential threat to the health, education, business, and tourism industries, collective mental health (MH) of populations, and execution of basic social and political rights. The combination of these developments has resulted in a global catastrophe that many governments did not anticipate, and which is likely to surpass any natural or human-caused disaster within the last decade. Furthermore, due to the complexity of the COVID-19 crisis, low- and middle-income countries were severely impacted, with their social, political, and economic systems buckling under the enormous stress of the pandemic. Countries employed various strategies including distributive leadership, far-reaching lockdown measures, curfews, and border closures to mitigate the deleterious effects of the pandemic [1].

Since the first case of COVID-19 in December 2019, frontline healthcare workers have been exposed to the risk of developing MH problems. Initial studies have also suggested a significant increase in the burden of care, impacting the psychological wellbeing of hospital employees [2,3,4,5,6,7]. Therefore, understanding the effect of a health crisis on the MH of hospital staff and being able to design and implement mitigatory interventions will help strengthen the healthcare system capacity and reduce the harmful psychological effects of the pandemic on the health workforce (HWF).

### 1.1. COVID-19 Response on a Small Island

Aruba is a small island developing state (SIDS) [8], with a gross domestic product (GDP) of 2693 million USD; it is a country within the Kingdom of the Netherlands. The island faces many challenges due to limited resources related to medical care and is vulnerable to natural disasters and international trade fluctuations. The healthcare needs are catered for by its major hospital Dr. Horacio E. Oduber Hospital (HOH) and an ambulatory healthcare facility, ImSan [9].

On 13 March 2020, the first case of COVID-19 was reported in Aruba. This number quickly rose to a peak of 63 cases within a short period, initiating national policy directives and targeted healthcare interventions to mitigate the spread of the virus. The decision to pre-emptively shut the island to foreign traffic was considered by many as an unequivocal and swift response to the pandemic. To aid the staff of the HOH, foreign medical staff (mainly from the USA) were recruited through a Dutch government initiative. The effects of these measures seemed helpful with a decline in transmission rates and number of deaths by 28 May 2020.

### 1.2. Economic Crisis during a Pandemic

In addition to the health consequences, the pandemic also had a deleterious impact on the island’s economy given its heavy reliance on tourism and vulnerable status as an SIDS [8,10]. While the authority’s swift response helped to curtail the human and economic damage, it could not avoid a severe GDP contraction [11]. As a result, the Aruban government was compelled to impose a salary cut for all public employees, including a 5% salary cut for HOH employees [12]. This decision is presumed to have significantly impacted the private lives of employees, the quality of life, and the relationship between the hospital management and staff. While the potential threats of the COVID-19 pandemic on the mental and physical health of the community have been reported, the impact of the economic crisis on healthcare professionals in the midst of a pandemic was not anticipated [3,4,5,6,7,13].

A recent review on the burden of COVID-19 on physicians’ MH [14] showed an alarming overview, with anxiety symptoms affecting 92.3% of physicians, and symptoms of PTSD in 75.2%. However, the unique intersection of economic and health(care) burdens on hospital staff within an SIDS setting is unknown. Hence, this study aims to evaluate the impact of COVID-19 as an unprecedented life event on the sense of wellbeing of HOH employees.

The objectives of the study were as follows:(a)To evaluate the impact of COVID-19, as an unprecedented life event, on the mental health of the hospital employees.(b)To provide recommendations and targeted interventions to mitigate the negative consequences on hospital employees’ mental health.

## 2. Material and Methods

### 2.1. Study Design

This mixed-method cohort study was designed to investigate the choice of specific psychological and social support strategies that HOH employees require in the face of the COVID-19 pandemic.

The mixed-method survey, designed by two physicians and two nurses, comprised a questionnaire survey (online) and multiple focus group interviews. The data from the questionnaire were collected between June and August 2020 via SurveyMonkey while six focus group interviews were conducted between August and October 2020.

For the questionnaire survey, participants were recruited through the hospital’s internal communication system. A memo explaining the purpose of the study and how participants could participate was placed on the hospital’s internal website. All employees were able to voluntarily access the anonymous online survey and provide their electronic informed consent before commencement. Additionally, heads of departments were given hard copies of the questionnaire for those who could not make use of the online medium. Our objective was to facilitate broad participation, by ensuring that access to the questionnaire was adapted to so that all employees could participate regardless of level of (digital) literacy. Participants were allowed to terminate the survey at any time desired.

### 2.2. Population

The study was a single-center descriptive survey, covering medical and nonmedical staff of the HOH. All staff members who were working during the first wave of the coronavirus disease outbreak were included. Participants included doctors, nurses, medical technicians, and other hospital staff.

### 2.3. Instruments: Quantitative Survey

A 25-item survey was designed comprising four parts and available in English, Papiamento, and Spanish. The first section of the survey consisted of six items measuring baseline sociodemographic information and medical history, as well as two additional items about perceived personal risk of COVID-19 in the workplace (four-point Likert scale, 0 = not at all to 3 = very much). The second section investigated the psychological impact of the pandemic and the healthcare needs of the participants. Using a four point-Likert scale, (0 = not at all to 3 = very much) the section contained four items that focused on the concerns and worries of hospital employees. Examples of such items included “What are your concerns as a healthcare worker if you have or should test positive for COVID-19”. The third part of the questionnaire investigated participants’ perceptions of the organizations preparedness to accommodate their safety and concerns while providing services to patients. This section consisted of five yes or no items and one item on a Likert scale (as in previous sections). Lastly, the fourth section of the questionnaire comprised five items that focused on the psychological impact of the pandemic on hospital employees. Subjective wellbeing (personal wellbeing index) and psychological distress measurement tools (Kessler psychological distress scale) described below were used for this purpose. The complete survey can be found in Appendix A. We tested the reliability of our instrument by examining the internal consistency of the relevant items in the questionnaire, which showed a high *Cronbach’s alpha* of 0.860.

### 2.4. Personal Wellbeing Index

This seven-item measurement provides global insight into perceived life quality, which is measured by items related to personal life satisfaction, on an 11-point Likert scale (0 = no satisfaction at all to 10 = completely satisfied) [15]. In the present sample, the index showed an internal consistency of 0.825 (Cronbach’s alpha).

### 2.5. Kessler Psychological Distress Scale

The Kessler Psychological Distress Scale screens for psychological distress, consisting of 10 items intended to yield a global measure of distress on the basis of items related to anxiety and depressive symptoms on a five-point Likert scale (1 = none of the time to 5 = all the time) [16]. In the present sample, the scale showed an internal consistency of 0.904 (Cronbach’s alpha).

### 2.6. Instruments: Qualitative Interviews

At the start of pandemic, the focus of the hospital’s initial response was on the technical aspect of organizing and providing good healthcare. However, it later became apparent that there was a need for psychological support among hospital employees and peers. A grassroots multidisciplinary team (later known as peer support network, PSN), consisting of three medical specialists, one social worker, an occupational social worker, and an administrative support staff) was formed, and subsequently endorsed by the hospital management. The team’s aim was to provide the hospital employees the opportunity to express and share their and experiences through focus groups on the various measures taken by the HOH leadership during the COVID-19 crisis. In August and October 2020, six group sessions were conducted, each with an average of six participants per session (range 3–10). Interpretative phenomenological analysis (IPA) was used for this arm of our investigation [17,18], which is a naturalistic strategy to explore how people give meaning to their personal and social worlds, and how they respond to a specific phenomenon [19,20]. There are three distinct features of IPA; it is idiographic because it focuses on the in-depth understanding of a phenomenon, in a small sample of individuals, [18,19,21], it is inductive in that it allows unexpected themes to emerge, and it is characterized by being interrogative. Consequently, IPA uses a phenomenological approach framed in an interpretative paradigm that aims to explore lived experiences ideographically and inductively. As a result, we could examine how participants perceived specific situations they encountered, to gain a richer picture of their lived experiences [18,19,21].

All group interviews were conducted in real time and documented by a scribe who attended each session. The transcripts were later used for thematic content analysis, to identify critical elements in the responses and allow comparison and categorization of the respondents’ perspectives [22]. We manually constructed the coding schemes used to identify the (sub)themes [23]. An inductive analysis process, with open codes, was used to evaluate the impact of the pandemic and its consequences on the general wellbeing of the employees [22]. Three researchers reviewed the (sub)themes separately; agreement on conflicting themes was achieved through consensus. The participants were provided with the opportunity to review the summaries of the group interviews and provide feedback on the data before final analysis.

### 2.7. Statistical Analysis

Data analysis, including calculating frequencies and proportions, was performed using SPSS statistics software. Associations were analyzed by Mann–Whitney (x^2^) test; a *p*-value < 0.05 was considered statistically significant.

### 2.8. Ethical Approval

Approval of the ethical committee was obtained by the HOH medical ethical committee.

## 3. Results

### 3.1. Quantitative Data

#### 3.1.1. Characteristics of Study Participants

All staff (*n* = 956) of HOH received the questionnaire; 28% filled out the questionnaire, among which 24% were completed and analyzed (Figure 1). Nurses filled out 117 questionnaires, doctors filled out 34, and 80 questionnaires were filled out by other HOH staff. The average age of those who completed the questionnaires was 41 years; 79% of those who filled in questionnaires identified as female, and the predominant profession was nursing with more than 5 years of experience. The high percentage of women is likely due to their high representation in the healthcare professions. In 2019, 70% of HWF members were women [24]. Few participants were single, and 53% of the population was otherwise healthy without comorbidities for a poor disease course of COVID-19 (Table 1).

#### 3.1.2. Factors Causing Concern

Seven personal factors (Figure 2) and nine work-related factors (Figure 3) were assessed with the use of the questionnaire: the working conditions, fear of getting infected, caring for infected patients, fear of infecting relatives, fear of stigma, fear of losing financial stability, PPE, test kits, staff shortage, adequate isolation space, guidelines in place, and training. In general, there was a light to moderate concern in the personal area after the first wave. Notably, employees were most concerned about infecting their relatives (mean 2.20, SD ± 1.0); however, this was not statistically significant. There was a clear fear of losing jobs and financial stability (mean 1.89, SD ± 2.00). Additionally, there was moderate concern about all work-related factors. The results were analyzed according to their profession (the analysis of all employee groups can be found in Appendix A). Nurses were significantly more concerned/anxious than doctors and other staff members on all accounts, about factors such as getting infected, being stigmatized, and financial stability. All groups were equally concerned that they might infect their relatives. Nurses were significantly more concerned about the availability of materials, isolation space, shortage of staff, training, and change in guidelines.

#### 3.1.3. Wellbeing and Psychological Health of Hospital Staff

The wellbeing of employees was screened through seven items on personal satisfaction (Figure 4). Items asking about future security scored poorest (mean 6.65, SD ± 2.1). The overall general wellbeing score indicated overall satisfaction with the current life and work environment of HOH employees. To assess the current MH status of staff, the employees were asked how often they felt tired, nervous, hopeless, restless, depressed, and worthless (Figure 5); their total response mean was 1.81 (SD ± 0.89). HOH employees scored higher on the level of tiredness (*p* = 0.271) and nervousness (*p* = 0.664), which can be interpreted as a likelihood of having a mild disorder.

#### 3.1.4. Coping Strategies and Support

Staff members were asked about their use of coping strategies during the first COVID-19 wave. One-hundred participants responded positively about the use of coping strategies to reduce stress during the COVID-19 pandemic (Figure 6). Respondents reported following strict protective measures (mean 2.26, SD ± 0.82) and learning about COVID-19 (mean 2.03, SD ± 0.92) as main coping strategies. Relaxing exercise in free time was another strategy to alleviate stress (mean 1.95, SD ± 0.88). Seeking professional help was not a commonly used coping mechanism (mean 0.24, SD ± 0.66). For future emergencies akin to the COVID-19 pandemic, the majority of staff did not want psychological support or were not sure if they needed any (Figure 7).

### 3.2. Qualitative Data

#### 3.2.1. Peer Support Network Interviews

The focus group responses were categorized by the following subjects: management, salary cuts, recruitment of external support staff, burnout, lack of recognition/satisfaction/understanding, quality of support, and quality of care (Table 2). In the initial interviews (August 2020), employees reported insufficient or no transparency in communication, leadership, unsafe work culture, and unnecessary hierarchy. Due to the influx of external staff, there was a perceived lack of equality between local and foreign employees. HOH employees often felt that their concerns were unheard or unrecognized, resulting in a generally low morale. Furthermore, the external staff required continuous training to be able to fit into the local working culture, the effect of which was a perennial perception of fatigue by the local team.

In the follow-up interviews (October 2020), the employees felt tired, felt demotivated, and did not feel that the HOH management served their expectations. They wished for more recognition for their efforts through (1) extra days off for employees working in COVID-19 departments, and (2) support in the form of a meal by the organization. The clash between foreign and local staff was still a cause for concern with some escalations since the original interviews. Due to the difference in training and care systems, the working-in process did not run smoothly, and HOH staff members were no longer willing to continue supervising newly trained personnel, who were moving through the system at a high rate. Due to growing exhaustion and frustration, as well as the extra time spent on training foreign staff, employees felt they could no longer guarantee the quality of care and training (compared to before the pandemic). They perceived an increase in patient-related medical complications on the wards due to high work pressure. Additionally, the news in the media about hospital staff did not match the reality of the situation at the bedside, resulting in increased tensions between staff and management.

#### 3.2.2. Reflexivity

Most of the authors of this study were employees of HOH, and we recognize that the relationship with the hospital may have some bias with respect to the analysis and interpretation of the data. For this reason, we reflected on how these relationships may have influenced the outcomes of this study. V.D. is a medical student from the Netherlands who came to HOH for 4 months to complete a clinical research internship. She did not have a previous relationship with the team or the hospital and, therefore, contributed some level of objectivity to the analysis and writing of the paper. J.d.K. is an infectious diseases specialist, and S.J. is a hospitalist. Their direct involvement in providing patient services at the frontline may have influenced some data interpretation, while providing important context to the significance of the findings. R.D. is a cardiologist, involved in several hospital committees at the time of the study. He was mainly involved with data analysis and interpretation and critically reviewing the manuscript for content and less with the conception and design of the study. We believe that his contribution added little or no bias to the interpretation of our findings. J.B. is a pediatrician and the chair of the hospital’s peer support network (PSN). It is possible that his role in the PSN could have been a potential bias in the design and implementation of the study. However, the role of the other members of the research team with no links to the network outweigh this. His role in PSN was helpful, however, in facilitating recruitment and engagement of hospital employees for the focused group interviews. Lastly, while there was an inherent risk of bias due to all of the authors’ roles in the hospital, we believe that the diversity of their specialties, as well as the differing lens through which they approached the pandemic, countered any bias that may have emerged as a result of these relationships.

## 4. Discussion

After 1 year of the pandemic, more studies are showing and supporting earlier claims that the burden on MH through isolation, quarantine, and fear evolves into long-term health problems and stigma [25], notably in high-risk populations. The current study assessed the impact of the first wave of the SARS-CoV-2 pandemic on the MH of the staff of HOH, examining the differences between employee groups, with the goal of providing recommendations for targeted support and coping strategies in future crises in an SIDS setting. This pandemic had a significant impact on HOH staff, through tiredness and anxiety, increased worries about job security, and staffing issues.

Previous studies [14] illustrated that all healthcare professionals responding to infectious outbreaks experienced increased levels of acute or post-traumatic stress and psychological stress. The questionnaire analysis demonstrated that HOH employees were moderately concerned in the personal domain during the first wave of COVID-19. The major concern was the fear of infecting their relatives, which reflects the local culture, where the family unit is the cornerstone of the Aruban community. Similarly, a study of medical staff in Hubei, China [26] found that staff often worried about their family, with the worry being highest in the age group most likely to have young families. This ties into an identified risk factor for psychiatric symptoms: having dependent children [27]. These worries were amplified in our study through significant financial instability of the island following the economic crisis [11] and supported by the survey responses and group interviews of the hospital employees. With the unprecedented physician burnout rates prior to the COVID-19 pandemic [28], this public health emergency also had a significant effect on the mental and professional wellbeing of the employers due to the (perceived) increased chance of infection, psychological symptoms, and their interaction [2,14].

Our study found an important level of concern with the high influx of foreign medical staff. There is little research into foreign medical aid during COVID-19; nevertheless, important questions were posed about the foreign aid in India during the case-surge of April 2021: “Will minimum stands of response be followed? Will the right capacities be recruited and scaled while considering the operational needs, ensuring that service delivery is implemented, monitored, and evaluated with transparency, accountability, and responsiveness to new challenges and needs?” [29]. These questions can be used to reflect on the responses of HOH staff. Despite the objective positive value of additional staff in the hospital, the presence of the foreign staff created tension among them, the hospital management, and the HOH employees, with the latter reporting fatigue and frustration, amplified by discrepancies in financial compensation between the external and local staff. Foreign aid can be of true value when used correctly, i.e., when the capacity, coordination, and resources are adapted to local needs [29].

The levels of tiredness and nervousness found in this study indicate mild psychological distress in HOH employees during the initial stages of the pandemic. These indicators can signal early stages of burnout in the hospital staff and can affect the delivery of patient care. Looking at other studies, MH problems are frequently encountered in medical staff during health crises and can have a lasting impact [27]. Healthcare workers were faced with an increased degree of stress, anxiety, depression, and insomnia due to the COVID-19 pandemic [6], with some reporting anxiety rates among HWF twice as often as prior to the pandemic [30]. Among hospital staff, nursing staff felt more anxious and nervous compared with other healthcare workers [26]; nurses were also more likely to develop PTSD and burnout [2], supporting the findings that nurses are significantly more worried about getting infected, stigmatization, financial stability, availability of materials and staff, training, and change in guidelines. This indicates that proximity to the patients carries an increased risk of psychological burden. Frequent epidemic-related dreams, reported by HOH staff, indicate the perceived trauma of this ongoing event [31] and are one of the core symptoms of PTSD. Psychological distress and perception of COVID-19 risk have a negative impact on hospital staff on the front lines [32] and add to the stress levels exacerbated in an emergency setting of a pandemic.

With regard to stress, several protective factors have been identified during health emergencies, namely, family safety [26], clear communication, emotional support from fellow staff [26,27], and social support [30]. Psychological interventions and positive coping styles [31] improve resilience and prevent anxiety, depression, and sleep problems [27]. HOH employees reported positive coping styles, such as following strict protective measures, learning about COVID-19, and relaxing exercises in free time, but not seeking professional psychological help. Those with negative coping styles had a poorer MH outcome [30]. Interestingly, our study found that majority of the HOH staff did not report a need for psychological support. This finding can be attributed to a cultural view of seeking professional support. Other research in the Caribbean islands has reported high levels of patriarchal and hierarchical structures [33,34], where seeking psychological help can be seen as a sign of weakness and is often surrounded by stigma. To respond to the perceived distress of employees, HOH set up the PSN, in tandem with this study. It revealed employees struggling with work-based financial remuneration, recruitment of external staff, and poor management communication during the first wave of the COVID-19 pandemic. Alternatively, clear communication from management could support and motivate staff [2,26], and psychological and education interventions could mitigate the negative effects of a health crisis [2,27]. The effect of PSN focus groups on HOH employees’ psychological wellbeing has not been assessed and should be further investigated.

Our findings illustrate the complexity of a health emergency, such as the COVID-19 pandemic, in an SIDS setting. Work-related frustrations can lead to higher levels of psychological distress in any setting, as corroborated by a study of frontline workers in the Netherlands [35]. A very recent study of the needs of healthcare workers in Singapore during the early months of COVID-19 [36] revealed a lack of effective communication, i.e., employees through feeling overwhelmed by conflicting messages and excessive information. The workers also felt they were not renumerated fairly for working overtime, and the workload distribution (particularly between junior and senior staff) exacerbated frustrations. Despite the vastly different setting of Singapore and Aruba, these issues in the HWF are in part reflected by our study. HOH staff faced the unpredictability of the first wave of the COVID-19 pandemic feeling frustrated by lack of appreciation (through salary cuts), and they faced the communication and management issues, with limited resources of a small island (amplified by an emerging economic crisis). While there are no studies investigating the HWF burden of the pandemic between high- and low-resource settings, there is room for interpretation of the burden on the individual staff as greater in comparison to a high resource setting.

## 5. Limitations

This cross-sectional study aimed to investigate the psychological wellbeing and the needs of the local hospital staff in a SIDS setting during the initial months of the COVID-19 pandemic and is the first such study. The survey was sent out to the entire hospital staff, aimed at obtaining an accurate portrayal of the wellbeing of the HWF in Aruba. With only 24% of employees analyzed, its representativeness of the whole institution could be questioned. While nurses comprise a considerable proportion of HOH employees and, as such, also the majority of the analyzed personnel, there may be an underrepresentation of the non-nurse workforce. Furthermore, since the surveys were anonymous, it is impossible to know if the participants of the focus groups had any overlap, or if it was a different group altogether. While this needs assessment focused on the local staff, the needs of the foreign staff were not investigated and warrant further research.

Key Recommendations for SIDS

Institutions should recognize the vulnerability of the small island setting and the importance of preparation before an acute situation.Communication is key between management and staff in an acute and long-lasting emergency.Organizations should introduce a set of clear guidelines for each step of the emergency/pandemic response which the staff can easily follow.Institutions should provide psychological support to the hospital staff by initiating peer support networks through external and impartial organization, and by implementing a task force to support the needs.

## 6. Conclusions

The people of Aruba experienced the economic impact of the global pandemic as more devastating, beyond the anticipated healthcare impact. Employees of HOH had work-related and personal concerns during the first wave of the COVID-19 pandemic. Those closest to the patients experienced higher concerns and emotional exhaustion than other professions.

Psychological and work-related support, teamwork, better communication information from hospital management, and job stability were among the preferred support modalities desired by employees.

Hence, comprehensive and holistic approaches should be implemented by the hospital management to prevent occupational burnout and demoralized work ethics and further emotional exhaustion.

## Figures and Tables

**Figure 1 healthcare-10-01263-f001:**
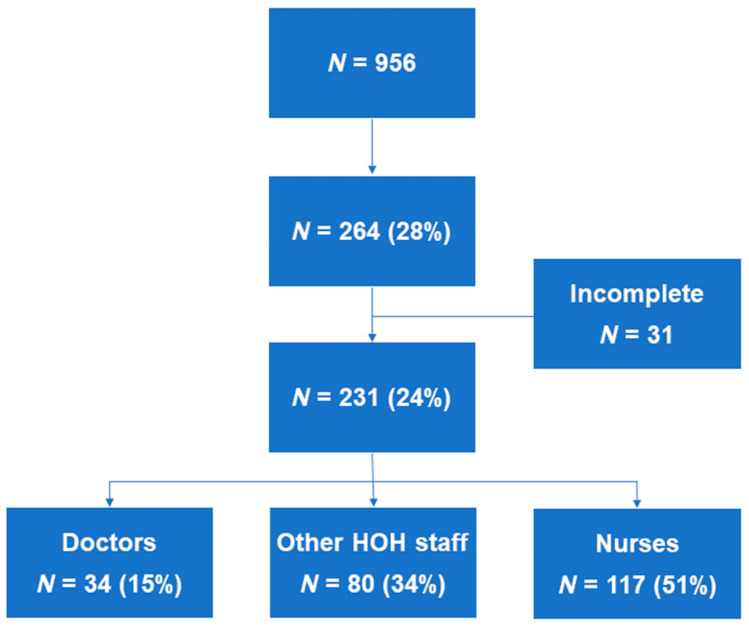
Flowchart of participants.

**Figure 2 healthcare-10-01263-f002:**
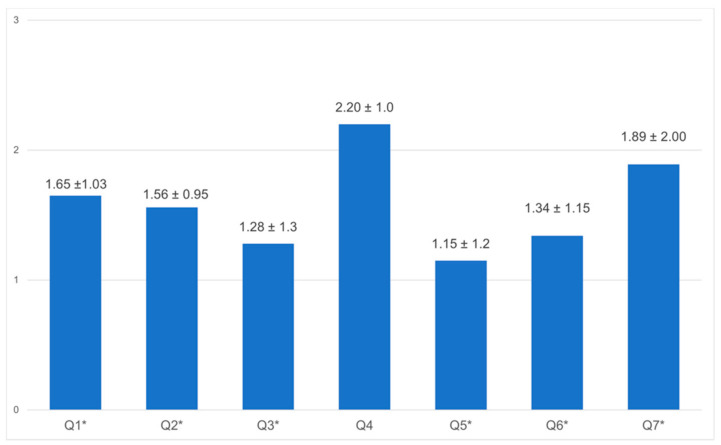
Staff feeling during COVID-19 outbreak concerning themselves, family, and financial stability. **Notes:** The items were scored from 0 = no concern to 3 = concern. A *p*-value ≤ 0.05 was considered statistically significant. Items: 1. Your profession as a healthcare worker makes you feel concerned; 2. Fear of getting infected; 3. You are concerned if you test positive, because you do not have self-quarantine space at home; 4. You are concerned if you test positive, because you might infect family members unintentionally; 5. You are concerned if you test positive, because of being stigmatized; 6.You are concerned if you test positive, because of fear of dying due to lack of specific treatment; 7. You are fearful of losing financial stability or job. * *p* ≤ 0.05.

**Figure 3 healthcare-10-01263-f003:**
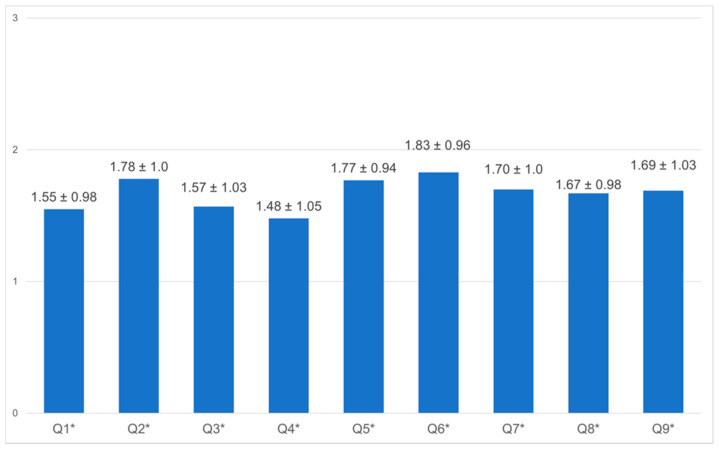
Overall feeling of HOH staff concerning hospital safety. **Notes:** The items were scored from 0 = no concern to 3 = concern. A *p*-value ≤ 0.05 was considered statistically significant. Items: 1. Your department or office makes you feel concerned; 2. You are concerned about the availability of personal protective equipment for HCWs; 3. You are concerned about caring for COVID-19 patients or persons under investigation for COVID-19; 4. You are concerned about the availability of testing kits and the duration of test results; 5. You are concerned about the number of staff available to take care of COVID-19 patients; 6. You are concerned about the availability of hospital beds/isolation units; 7. You are concerned about the adequate supply of cleaning and sanitation materials. 8. You are concerned about the fast-changing guidelines for managing patients. * *p* ≤ 0.05.

**Figure 4 healthcare-10-01263-f004:**
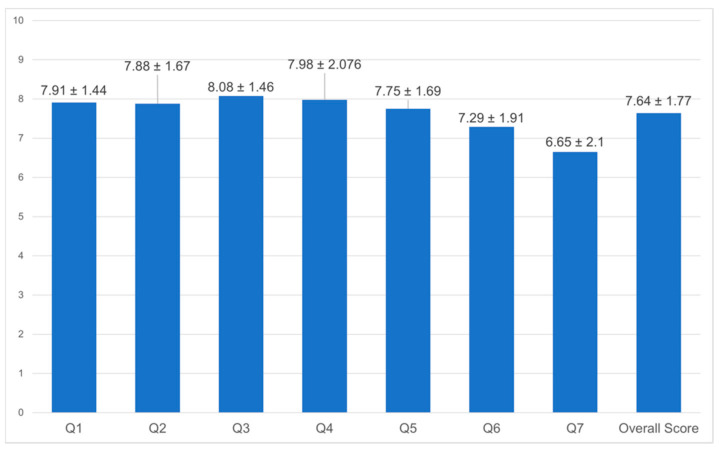
General wellbeing of HOH staff. **Notes:** Personal wellbeing scale: 0–10; 0 = no satisfaction at all, 10 = completely satisfied. Items: 1. How satisfied have you been with your standard living? 2. How satisfied have you been with your health? 3. How satisfied have you been with what you are achieving in life? 4. How satisfied have you been with your personal relationships? 5. How satisfied have you been with how safe you feel? 6. How satisfied have you been with feeling part of the community? 7. How satisfied have you been with your future security? *p* ≤ 0.05.

**Figure 5 healthcare-10-01263-f005:**
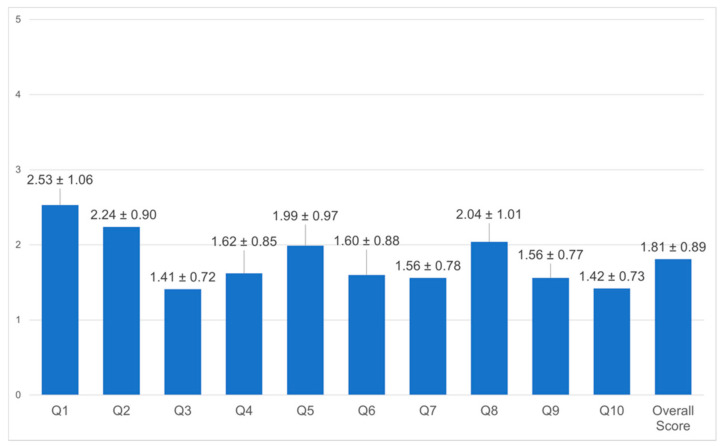
Psychological distress level in the past 4 weeks. **Notes:** Kessler psychological distress scale of 1–5, where 1 = never, 5 = always. Interpreting results: 1–1.9 likely to be well, 2–2.5 likely to have mild disorder, 25–29 likely to have a moderate disorder, 30–50 likely to have a severe disorder. Items: 1. How often do you feel tired? 2. How often do you feel nervous? 3. How often do you feel so nervous that nothing can calm you down? 4. How often do you feel hopeless? 5. How often do you feel restless and fidgety? 6. How often do you feel so restless that you cannot sit still? 7. How often do you feel depressed? 8. How often do you feel that everything is an effort? 9. How often do you feel so sad that nothing can cheer you up? 10. How often do you feel worthless? *p* ≤ 0.05.

**Figure 6 healthcare-10-01263-f006:**
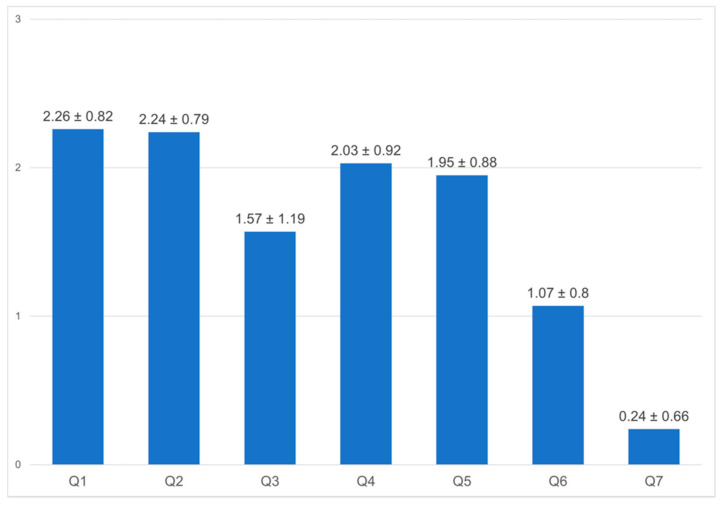
Personal coping strategies used to alleviate stress during COVID-19 pandemic. **Notes:** Participants who reported that they used coping strategies were asked to answer additional items on a scale of 0–3, where 0 = never, 1 = sometimes, 2 = often, and 3 = always. Items: 1. Followed strict protective measures such as hand washing, masks, face masks, and protective clothing; 2. Learned about COVID-19, its prevention, and its mechanism of transmission; 3. Chose not to be in the hospital unnecessarily; 4. Chose to avoid public places; 5. Did some relaxation activities in free time; 6. Avoided media news about COVID-19 and related fatalities; 7. Sought help from a professional. *p* ≤ 0.05.

**Figure 7 healthcare-10-01263-f007:**
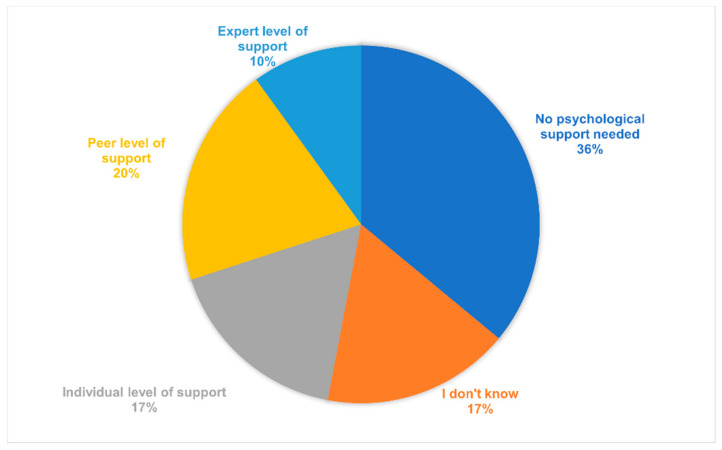
Preferred support needed at work for future outbreaks. **Notes:** Reponses of participants about their preferred types of support. Individual-level support: access to self-help resources directly online or in paper format. Peer-level support: support provided by peers to individuals or groups. Expert-level support: provision of professional help, e.g., psychiatrist, psychologist, or social worker. The number of recorded responses is shown in parentheses. The final item allowed participants to freely comment on what they thought they needed from their employer. Their responses were matched and clustered into distinct categories. The three most mentioned categories were teamwork and financial stability, better communication from the hospital leadership, and financial and work stability (*n =* 24, 18, and 20, respectively).

**Table 1 healthcare-10-01263-t001:** Staff demographics.

Characteristics	Total = 231
Age, Years (mean SD)	41 (11)
Gender, *N* (%)	
Female	183 (79)
Male	48 (21)
Occupation, *N* (%)	
Medical	170 (74)
Nonmedical	61 (26)
Profession, *N* (%)	
Nurse	117 (51)
Doctor	34 (15)
Other hospital staff	80 (34)
Years in occupation, *N* (%)	
<1 year	29 (13)
1–5 years	73 (32)
>5 years	129 (56)
Marital status, *N* (%)	
Single	31 (13)
Partner/spouse/children/parents	200 (87)
Cumulative Risk factors, *N* (%)	
0	122 (53)
1	55 (24)
2	17 (7)
3	8 (4)
4	15 (6)
>5	14 (6)

**Table 2 healthcare-10-01263-t002:** Responses from group interviews during the peer support network meetings.

August 2020	
Management	There is insufficient communication and a lack of transparency about what is going on in management. Many people experience the culture as hierarchical and unsafe. They do not dare (anymore), they do not feel like it (anymore), it does not matter anyway, or they do not feel heard (anymore). Clinical laboratory employees do not always receive information from the hospital and are not well informed.
Salary cuts	Employees want to show solidarity, but a 5% cut in salary has an impact on the private lives of some, who are not given a chance to voice their concerns about it. There is a need for clarification about what will happen with the 5% cut in salaries. The decision is perceived as a top-down one, with insufficient explanation about the reason for the decision, and employees’ personal situations are not considered. Will there be some form of compensation, for example, in the form of days off, as compensation for the 5% withheld from salaries?
External staff	It seems as if external staff are compensated more than permanent employees with the same tasks/roles. This creates a sense of inequality. The local staff members feel that they had to put in twice as much effort with the presence of the external staff that they had help fit in. Local staff members get nothing in return for their efforts in assisting the external staff, except directions to pay closer attention to them. Due to differences in training between the external and local staff, there is a difference in professional performance and knowledge of local practices and equipment. Hence, external staff required more help than usual from the local team. A lot of investment is needed to train the external staff to work in the hospital. In addition, they often leave just after successfully being trained, resulting in a high turnover in training process. Local hospital staff feel that a lot is imposed on them and that they can do nothing about it: too little input from the work floor. Very high absenteeism was experienced.
**October 2020**	
Exhaustion (Burnout)	The greatest frustration of employees is the continuous training of new and temporary employees coming from abroad. People experience that the identity and overview of their department has been lost, resulting in a loss in the pleasure of work. Many feel that they cannot take a rest and often work overtime to make the work manageable for the staff in next shift. There is detachment from work, job dissatisfaction, and a lack of joy in work. There is the feeling of having to come to work because they are obliged to do so. Employees experience that the care for COVID-19 patients is physically demanding. They believe that they need extra days off to recuperate after working on the COVID-19 wards. Employees start to dream about work and still experience high levels of stress when they go home. The staff feels very responsible for keeping the COVID-19 patients alive.
Lack of recognition/satisfaction/understanding	Employees want recognition for their efforts. They experience their relationship with the employer as unidirectional. They indicate that the cut in their salary has made working conditions worse; single parents feel that they are working hard and afterward do not have enough for their own family at home. Employees experience that some specialist physicians expected everything to return to normal as soon as possible without considering the consequences. There is a lack of mutual interests, resulting in a lot of stress. Employees indicate that there are some shifts where they can hardly find time to eat or get food. They feel it would be thoughtful if the organization could help to provide healthy snacks or meals during busy shifts. Many leave work with a sense of dissatisfaction or feeling that they have not achieved enough. The sense of belonging among the local staff is absent with the external staff. This creates a feeling that the department is collapsing and affecting daily operations. Many are tired, are demotivated, and feel misrepresented in the mainstream news and social media.
Quality of Support	The external staff members are experienced professionals, but their help is not what the staff expected. For example, some external staff do not know how to work with local equipment, e.g., mechanical ventilation systems, or lack the experience to perform certain procedures which is tasked out to other professional such as ventilation technicians. The high turnover is experienced as demotivating when it comes to training new staff. Many local staff feel tired and frustrated, resulting in conflicts and escalations between colleagues
Quality of Care	Local staff members feel that medical complications such as pressure ulcers in hospitalized patients occur more frequently than before. Employees indicate that situations are unsafe. They do not feel heard (anymore) by the hospital management and experience that the situation is imposed on them, and that they are not given any choice in the matter. Employees feel that they cannot provide the quality of care which they normally do, due to busyness in departments and training of new external staff. As much of their focus is diverted to training others, the quality of care cannot be guaranteed. It is impossible to keep track of the number of foreign employees, because various processes do not run as they should. People are trained but do not keep to agreements; permanent staff members are tired and no longer willing to correct this behavior. Students and trainees feel that there was a decrease in the quality of care in the departments, below what it was before the crisis, thus affecting the quality of their training. They feel that they often take on professional responsibilities that they are not yet qualified to do.

**Notes:** A total of six sessions were conducted in August and October 2020 by the Peer Support Network.

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
