# Peer review of "Managing the Mental Health of Healthcare Professionals in Times of Crisis: The Aruban COVID-19 Experience"

_healthcare, 2022, doi:10.3390/healthcare10071263_

Round 1

Reviewer 1 Report

Dear Authors,

thank you for the possibility of reading your paper.

This is a valuable study on the perception of the impact of the coronavirus (COVID-19) pandemic among medical personnel in Aruba.

This study involves both qualitative and quantitative methods and this is a strong point of this paper. 

However, there are some issues that should be covered.

1. I would be careful with using the word "impact" as this is a cross-sectional study. I would propose to use "perception of the impact of the COVID-19 pandemic" or "experience of the COVID-19 pandemic" to emphasize the subjectivity.

2. Please, describe thoroughly the research methods with the exact number of items, Likert scale, and reliability in this study. The measurements that are presented for the first time should include also factor analysis and external reliability.

Author Response

Kindly find our response to the reviewer's comments in the attachment

Reviewer 2 Report

This manuscript's overall readability, writing quality, and relevance are good. The authors did an excellent job providing a contextual basis for the study and its utility for healthcare organizations, workers, policymakers, and the general public.  

While I see the article as publishable and easily improved, I offer several suggestions for improving the manuscript. Next, I have an overriding concern related to the study's trustworthiness.

A few minor suggestions to clarify some points in the manuscript follow. 

SUGGESTIONS:

The author's choice of words and details added depth and clarity to the importance of the study. However, there were some areas of the manuscript where the wording was less explicit and encouraged assumption-making.  

Page 2, Paragraph 1: 

The authors refer to the study's value in "strengthening the healthcare system capacity ." I believe the word capacity requires elaboration to ensure its intended meaning.

Page 2 Next to the last Paragraph before Materials & Methods:

The authors refer to the study's aim "to evaluate the impact of COVID-19 as an unprecedented life event on the sense of well-being of HOH employees". Although COVID-19 is named an unprecedented life event, there is the potential suggestion that HOH employees have not experienced similar challenges with infectious disease crises. 

Page 3, second Paragraph:

The authors indicate that "All HOH workers were requested to voluntarily participate in this anonymous online survey distributed through hard copy, emails, WhatsApp, and other communication tools ." This statement reveals more about how the surveys were accessed and less about the recruitment method for participation. 

Some speculation about the 24% response rate might be helpful in the interpretation of the results. 

INTERESTING THAT THE STAFF IDENTIFIED NOT WANTING PSYCHOLOGICAL SUPPORT. Some discussion of this finding seems warranted. 

Page 2, first Paragraph, the authors referenced an article about the mental health risks to healthcare workers since the first case of COVID-19 in December 2019." Despite the authors using a citation, the reader may confuse the reference to the emerging pandemic across the continents with the specific problems posed by the pandemic at the HOH.

While these suggestions are minor, I think they will add clarity for a broader audience. 

More salient to the manuscript's publication is making the underlying questions about potential bias more transparent.  

TRUSTWORTHINESS OF THE STUDY: 

The authors' 25-item questionnaire designed for the study does not provide sample questions nor reliability or validity measures.

The authors indicate they considered 80% of questions answered acceptable for inclusion in the analysis. I would be interested to know more about what kinds of questions were omitted by participants. Were there any criteria applied regarding which questions were critical to being answered for inclusion? 

I was also confused by the authors asserting that they conducted in-depth qualitative interviews. It appeared that the qualitative arm of the study was focus groups as opposed to in-depth qualitative interviews. The authors did not state the specific aim of the focus groups nor detail how the focus groups were derived or conducted. 

It is unclear if the focus group participants were the same as those who participated in the survey. If so, how was the anonymity of survey responses maintained? 

There were two areas of the focus groups that I found misleading: 

  • "Following the initial quantitative survey, a follow-up in-depth qualitative study was conducted to investigate and better understand the employees' lived experiences.
  • Following the COVID-19 crisis, a Peer Support Network (PSN) was set up to provide hospital employees the opportunity to express and share their concerns and experiences on specific measures taken by HOH during the COVID-19 crisis."

It is unclear if the PSN served as an intervention for staff stress and as the source of participant selection for focus groups. 

If the surveys were anonymous, it would seem improbable that survey and focus group participants were necessarily the same. Thus, the analysis and comparison of respective responses in each data source would need to be interpreted with that caution in mind. 

The authors state the study was designed "to investigate the choice of specific psychological and social support strategies that HOH employees require in the face of the COVID-19 pandemic". However, the conclusions focus on the hospital's need to implement "comprehensive strategies and holistic approaches to prevent burnout and demoralization." 

The study's design raises questions about whether it arose from an internal system-driven response by HOH instead of a scholarly investigation of staff responses to COVID -19 as the phenomenon to be explored.

The staff's salary cuts appear likely to conflate any other issues emerging from the data. It is unclear how much this issue was explored and considered in interpreting findings. 

For example, the authors' commentary on workers' dissatisfaction with work burden focused more on the foreign aid of other healthcare workers. It failed to explore how the angst toward management intersects with other issues emerging in the day-to-day stress of the pandemic.

Lastly, I would like to see some disclosures from the authors, particularly regarding their relationship with the hospital.

Specifically, it would add to the study's rigor if the authors addressed how they perceived their positions may have affected the study's design, participation rates and responses in surveys and focus groups, and the conclusions about the data. 

By explicitly addressing these issues, this otherwise well-written manuscript, in my opinion, would create a more scholarly presentation of the research.   

Author Response

(The authors gave the same response as above.)

Reviewer 3 Report

Thank you for the opportunity to review this study entitled “Managing the mental health of healthcare professionals in times of crisis: The Aruban Covid19 experience.” (healthcare-1766708).

The study focused on the psychological effect of the COVID-19 pandemic, by investigating the impact of the first wave of the SARS-CoV-2 pandemic on the mental health of staff.

In my opinion, the research topic is relevant, and the study is interesting. Parallelly, there are some issues that need to be addressed before the paper will be suitable for publication.

1.     Abstract: the information about the sample should be deepened (Mean age and SD? Percentage of men and women?) to provide a clear picture of what will be presented in the paper.

2.     Introduction: In my opinion, it would be good to refer to trend or longitudinal studies, if any. Since the authors frame this study considering the impact that COVID-19 has on a psychological level, I suggest some research to propose a comprehensive framework in the introduction, which should be supplemented with further literature search by the authors:

- Hyland et al., 2021; doi: 10.1016/j.psychres.2021.113905.

- Gori & Topino, 2021; doi: 10.3390/ijerph18115651

- Wang et al., 2020; doi: 10.1016/j.bbi.2020.04.028

To find the suggested articles, the authors can use this source: https://www.doi.org/

3.     Please, report the internal consistency (Alpha or Omega values) of the Kessler Psychological Distress Scale in the present sample.

4.     The gender imbalance should be highlighted as a limitation.

Author Response

(The authors gave the same response as above.)

Round 2

Reviewer 1 Report

Thank you for providing changes.